# Studies on the Efficacy, Potential Cardiotoxicity and Monkey Pharmacokinetics of GLP-26 as a Potent Hepatitis B Virus Capsid Assembly Modulator

**DOI:** 10.3390/v13010114

**Published:** 2021-01-15

**Authors:** Selwyn J. Hurwitz, Noreen McBrearty, Alla Arzumanyan, Eugene Bichenkov, Sijia Tao, Leda Bassit, Zhe Chen, James J. Kohler, Franck Amblard, Mark A. Feitelson, Raymond F. Schinazi

**Affiliations:** 1Center for AIDS Research, Laboratory of Biochemical Pharmacology, Department of Pediatrics, Emory University School of Medicine and Children’s Healthcare of Atlanta, 1760 Haygood Drive, Atlanta, GA 30322, USA; shurwit@emory.edu (S.J.H.); sijia.tao@emory.edu (S.T.); lbassit@emory.edu (L.B.); zhe.chen@emory.edu (Z.C.); jjkohle@emory.edu (J.J.K.); famblar@emory.edu (F.A.); 2Department of Biology, College of Science and Technology, Temple University, Philadelphia, PA 19122, USA; Noreen.McBrearty@temple.edu (N.M.); alla.arzumanyan@temple.edu (A.A.); evgbic@temple.edu (E.B.); feitelso@emory.edu (M.A.F.)

**Keywords:** GLP-26, hepatitis B virus, capsid effector, cardiomyocytes, cynomolgus monkeys, pharmacokinetics, oral bioavailability, mouse model

## Abstract

While treatment options are available for hepatitis B virus (HBV), there is currently no cure. Anti-HBV nucleoside analogs and interferon-alpha 2b rarely clear HBV covalently closed circular DNA (cccDNA), requiring lifelong treatment. Recently, we identified GLP-26, a glyoxamide derivative which modulates HBV capsid assembly. The impact of GLP-26 on viral replication and integrated DNA was assessed in an HBV nude mouse model bearing HBV transfected AD38 xenografts. At day 45 post-infection, GLP-26 reduced HBV titers by 2.3–3 log_10_ versus infected placebo-treated mice. Combination therapy with GLP-26 and entecavir reduced HBV log_10_ titers by 4.6-fold versus placebo. Next, we examined the pharmacokinetics (PK) in cynomolgus monkeys administered GLP-26 via IV (1 mg/kg) or PO (5 mg/kg). GLP-26 was found to have 34% oral bioavailability, with a mean input time of 3.17 h. The oral dose produced a mean peak plasma concentration of 380.7 ng/mL, observed 0.67 h after administration (~30-fold > in vitro EC_90_ corrected for protein binding), with a mean terminal elimination half-life of 2.4 h and a mean area under the plasma concentration versus time curve of 1660 ng·hr/mL. GLP-26 was 86.7% bound in monkey plasma. Lastly, GLP-26 demonstrated a favorable toxicity profile confirmed in primary human cardiomyocytes. Thus, GLP-26 warrants further preclinical development as an add on to treatment for HBV infection.

## 1. Introduction

Hepatitis B virus (HBV) causes significant morbidity and mortality globally [1,2]. Among some HBV carriers, this is also associated with the development of chronic hepatitis, which can progress to fibrosis, cirrhosis and hepatocellular carcinoma (HCC) [3]. Several vaccines exist, but they are ineffective in individuals with established infections. Current antiviral nucleoside analogs (NUCs) and interferon therapies rarely clear covalently closed circular DNA (cccDNA), requiring lifelong treatment [4]. NUCs in their 5′-triphosphate form inhibit viral polymerase by acting as chain terminators, thus blocking elongation of the viral DNA strand. However, HBV persists in the majority of chronically infected individuals due to the formation of cccDNA in the nucleus of hepatocytes. Thus, current therapies require long-term administration to maintain suppression, which may result in reduced compliance and the emergence of resistant virus [3].

Accordingly, there have been major efforts to develop direct-acting HBV compounds with other mechanism(s) of action [5]. Since HBV replication occurs within immature core particles, the assembly and stability of these core particles is a potential target for developmental therapeutics [6]. In recent years, various HBV capsid assembly modulators (CAMs) have been developed which inhibit viral assembly and/or compromise the stability of these core particles or their entry into the nucleus [7]. Since CAMs and NUCs have different mechanisms of action, combination regimens may enhance antiviral efficacy, while reducing the likelihood of developing drug resistance [8].

Our group recently published on GLP-26, a novel non-toxic glyoxamide derivative which modulates HBV capsid assembly with low nanomolar antiviral activity, resulting in reduced cccDNA and HBV antigen levels in vitro [9]. Strikingly, in a humanized HBV mouse model, combination treatment with entecavir (ETV) decreased viral loads and viral antigens (such as HBeAg and HBsAg) with sustained virology response (SVR) for up to 12 weeks after treatment cessation [9]. Based on these results, further in vivo characterization and preclinical development were warranted. We examined the anti-HBV efficacy of GLP-26 and ETV in nude mice supporting the growth of subcutaneous human hepatoblastoma cells replicating HBV (AD38 cells) [10] previously established by Feitelson et al. and compared the results with those from the humanized mouse study [9,11]. In this model, nude mice bearing AD38 tumor xenografts stably transfected with HBV and under the control of a tetracycline (TET) operon were used [11,12]. In the presence of TET, TET repressor expression is stimulated, thereby inhibiting the transcription of pre-genomic HBV RNA, which is then reverse transcribed into progeny HBV DNA during virus replication. In the absence of TET, high levels of virus replication are obtained. Since drug-induced cardiotoxicity is one of the major side effects associated with drug development and is one of the leading causes of clinical trial discontinuation, these studies included an in vitro evaluation of potential toxicity in cardiomyocytes [13,14]. Further, a single-dose study was performed in male cynomolgus monkeys following intravenous (IV) and oral (PO) administration to assess the plasma pharmacokinetics (PK) of GLP-26.

## 2. Materials and Methods

### 2.1. Chemicals

GLP-26 (molecular weight = 373.4 g/mol), BMS-986094 and emtricitabine (FTC) were prepared in house according to published procedures [9]. The purity of these compounds was greater than 98% as determined by high-performance liquid chromatography (HPLC) and NMR methods. Verapamil (molecular weight = 454.6), lopinavir (molecular weight = 628.8) and FTC (molecular weight = 247.2 g/mol), used as assay reference standards, were purchased from Sigma Chemicals. (St. Louis, MO, USA) or provided from our library of antiviral compounds. Acetonitrile (HPLC grade), dimethylsulfoxide (DMSO), polyethylene glycol 400 (PEG400), and all other chemicals (analytical grade) were purchased from Fisher Scientific (Fair Lawn, NJ, USA).

### 2.2. Evaluation of GLP-26 in AD38/Nude Mice

A nude mouse/AD38 mouse model was used to characterize the antiviral activity of GLP-26 alone and in combination with ETV [11,12]. Studies were approved by the Institutional Animal Care and Use Committee (IACUC) of Temple University (Animal Protocol No. 4954, issued April 14, 2018). Briefly, young male nude mice were subcutaneously injected with 1 × 10^7^ HepG2/AD38 cells in a single location on their backs [10]. When tumors reached 5 mm diameter 2–3 weeks later, mice were retro-orbitally bled and the baseline serum HBV titers were measured by real-time PCR [11] exactly as described [11]. Tumor-bearing mice were then administered tetracycline (TET) in their drinking water (2.5 mg/mL, refreshed every other day) for 10 days to allow additional tumor growth while virus replication remained suppressed. Mice were then taken off TET, bled again to determine HBV titers, and treated for another 45 days with ETV (10 mg/kg, 3 times per week), GLP-26 (55 mg/kg, 3 times per week), combination therapy (same dosing and schedule), or placebo (sucralose only) via feeding tubes on Mondays, Wednesdays and Fridays. Virus titers were assayed by PCR, and tumor sizes were measured by caliper, on days 20, 30 and 45 after the removal of TET. HBV titers (T = HBV DNA copies/mL serum) were divided by tumor volume (in mm^3^). V was calculated using the formula V = (a × b^2^)/2, equal to (width × length squared) divided by 2 [15]. The ratios of T/V (HBV counts/mL serum per mm^3^ tumor), were plotted over time (Figure 1). Tumors were removed at the end of the experiment, sectioned and examined for visible and microscopic signs of necrosis. Based on previous experience, the experiment was terminated at 45 days in order to avoid ulceration that would have occurred if the experiment was extended further. Use of these mice in this experimental protocol above was approved by the Temple University IACUC.

### 2.3. Measuring the Unbound Fraction of GLP-26 in Cynomolgus Monkey Plasma

It was assumed that unbound fraction of GLP-26 in plasma is related to antiviral efficacy [16] The unbound fraction of GLP-26 was measured in cynomolgus monkey (1 donor) and human (6 donors pooled) plasma and in 2% bovine serum (FBS), utilizing a Centrifree Ultrafiltration Device with an Ultracel PL membrane, following the instructions included by the manufacturer (Millipore Sigma, St. Louis, MO, USA). For comparison, lopinavir (positive control, reported to be ~98 to 99% bound in human plasma) or emtricitabine (negative control, reported to be <4% bound) were analyzed in spiked human plasma in the same experimental run [17,18]. Plasma and 2% FBS samples were spiked with 10 µM of a test compound, placed in the tube above the membrane and the tubes were centrifuged. The plasma-free filtrates (lower layers) were then harvested and assayed by LC–MS/MS. The non-specific adsorption for GLP-26 in the Centrifree ultrafiltration device was evaluated before the plasma binding assay, with low adsorption (approximately 2.6%). Therefore, percent free compound was calculated as µM concentration in the lower layer/10 µM × 100. By extension, the percent of compound bound to plasma equals 100—percent free compound.

### 2.4. Pharmacokinetic Studies with Male Cynomolgus Monkeys

The pharmacokinetics of GLP-26 was assessed in six male, non-naïve cynomolgus monkeys from the Covance colony (Covance Research Products, Alice, TX, USA). Studies were approved by the Institutional Animal Care and Use Committee (IACUC) of Covance Laboratories, Inc. (Assurance No. A3218-01, issued 12 June 2018). Male monkeys (2.8–3.7 kg) were administered single doses of GLP-26 via IV (1 mg/kg infused in 0.33 mL/kg over 30 min, n = 3) or PO (gastric gavage, 5 mg/kg, as a 1.67 mL/kg suspension, n = 3) in a vehicle containing 20% dimethylsulfoxide (DMSO), 50% polyethylene glycol 400 (PEG400) and 30% normal saline. No anesthesia was used. Serial blood samples (0.5 mL/sample) were taken at 0.25, 0.48 (before end of infusion), 0.58, 0.75, 1, 1.5, 2, 4, 8, 12, 24, and 48 h post-dose (based on the time of oral administration or start of infusion). Blood samples were centrifuged and plasma collected, kept on ice, and frozen at −70 °C until used for analyses. GLP-26 concentrations were assayed by Covance Laboratories Inc. (Madison, WI) with a generic liquid chromatography system with tandem mass spectrometric detection (LC–MS/MS), with verapamil as an internal standard. Analyst^®^ software (Version 1.6.2) was used to capture the LC–MS/MS data and integrate the peak areas. Watson LIMS software (Version 7.4.1) was used for data storage, management and reporting. The limit of quantification (LOQ) of the assay was 1 ng of GLP-26 per mL of plasma.

#### PK Analysis

Non-compartmental PK analysis was performed in our laboratory on the plasma concentration versus time profile of each monkey using the Kinetica^TM^ program (5.1, Thermo Scientific, Tewsbury, MA, USA), and graphs were plotted using the ggplot2 package in the R program (R version 3.6.1; R Statistical Foundation, Vienna, Austria http://www.r-project.org/) [19]. GLP-26 concentrations below LOQ could not be analyzed. The resulting PK parameters (means and % coefficient of variation (%CV)) are summarized in Table 1. Maximal plasma concentrations GLP-26 (C_max_) and corresponding times (T_max_) were observed values. Areas under the plasma concentration versus time curves (AUC_total_) were computed using the “arithmetic up”, “log down” method, with extrapolation. Other parameters reported included clearance (CL/F = dose/AUC, L·hr^−1^·kg^−1^), terminal first-order half-life observed in plasma (t_1/2_, h^−1^), area under the first moment curve (AUMC, ng/mL·hr^2^), mean residence time (MRT = AUMC/AUC, h) and apparent steady-state volume of distribution (V_ss_/F). The fraction of the oral dose absorbed (*F*) was computed using the formula *F* = 100 × (AUC_oral_/dose_oral_)/(AUC_IV_/dose_IV_), where AUC_oral_ and AUC_IV_ are the extrapolated AUCs after oral and IV dosing, respectively (assumes F of the IV dose = 1). The mean input time of the oral dose into plasma (MIT) = MRT_oral_—MRT_IV_, where MRT_oral_ and MRT_IV_ represent the mean residence times extrapolated to infinity after oral and IV dosing, respectively [20].

### 2.5. Effect of GLP-26 on Human Cardiomyocyte Function

Freshly thawed cryopreserved human cardiomyocytes were seeded in 24-well collagen-coated plates with iCell plating medium at a density of 240,000 (plating efficiency 53%) and incubated in a cell culture incubator at 37 °C, 7% CO_2_. After 48 h, non-adhered cardiomyocytes and debris were removed by rinsing twice with iCell maintenance medium and the attached cells were incubated for an additional eight days in maintenance medium, with fresh medium replacement every other day. Ten days post-plating when all cells demonstrated regular synchronous beating, cardiomyocytes were then exposed to 2 and 10 µM (similar to the maximal total (free + bound) plasma concentration) of GLP-26 prepared in DMSO. The same volume of DMSO was used as negative control, and 50 µM of BMS-986094 was used as positive control [21]. Cultures were imaged using a PerkinElmer spinning disc confocal microscope (PerkinElmer, Waltham, MA, USA) and 30 s videos were recorded at 14 fps using a Hamamatsu Flash 4.0 sCMOS camera and Velocity Software at 2, 4, 8, 12, 24, and 48 h after drug exposure. Automated video-based analysis of contractility by a computational motion tracking software was applied to identify potential cardiac effects of GLP-26 [22].

#### Statistical Analysis

Comparison of virus titers in treated and control nude mice replicating HBV from AD38 cells was evaluated by the Student’s *t*-test.

## 3. Results

### 3.1. Effect of Antiviral Monotherapy or Combination Therapy on HBV DNA Copies/mL Serum in Nude Mice Bearing HBV-Infected AD38 Xenografts

Within each experimental group, mice with larger HBV transfected AD38 tumors demonstrated higher HBV titers in their sera. Therefore, serum HBV titers (T, DNA copies/mL) were normalized by AD38 tumor volume (V, mm^3^) and the resulting T/V ratios (HBV DNA/mL serum per mm^3^ tumor) were plotted versus time (Figure 1).

Group I (eight nude AD38 xenograft) mice were treated with ETV monotherapy (10 mg/kg on Mondays, Wednesdays and Fridays, via feeding tube) for 45 days. Seven out of the eight mice showed a decrease in HBV DNA levels (Figure 1A). There was a mean 3.7-fold (0.562 log_10_) reduction in T/V ratios between days 0 and 45 and T/V ratios did not rebound during ETV treatment (*p* > 0.4). One mouse demonstrated a modest rebound of 1.28-fold by day 45. Similarly, most mice in group II treated with GLP-26 alone (55 mg/kg on Mondays, Wednesdays and Fridays via feeding tube) also had reduced T/V ratios, but only partially, and T/V ratios increased over time to near pre-treatment levels (Figure 1B). Eight of the ten mice had a mean 13.8-fold (1.14 log_10_) rebound in T/V and there was no evidence of rebound in the remaining two. However, a comparison of T/V ratios between days 0 and 45 did not demonstrate statistical significance (*p* > 0.4), suggesting that GLP-26 continued to partially suppress virus replication. All nine mice in group III co-administered GLP-26 and ETV (at the same ETV and GLP-26 doses as in monotherapies), showed a greater than 111-fold (2.05 log_10_) mean decrease in T/V ratios by day 45 of treatment (Figure 1C). Most notably, combination therapy was more efficacious than either monotherapy, and in five of nine mice, serum HBV DNA became undetectable by qPCR by day 20 of treatment and remained undetectable (or at very low levels) for the remainder of the experiment. Once combination therapy reduced virus titer to undetectable, no rebound was observed, provided combination therapy was continued through the duration of the experiment (day 45). In contrast, when combination therapy was stopped on day 35, rebound was observed in some mice. Specifically, all seven mice taken off TET after 10 days, and left untreated for the remainder of the experiment, showed an average of 340-fold increase (2.53 log_10_) in T/V ratios by day 45 (Figure 1D).

Comparison of mean T/V ratios of all the treated mice (from Figure 1A–C) versus control mice (from Figure 1D) revealed a consistent suppression of virus replication in all drug treatment groups compared to mice not administered GLP-26 or ETV (Figure 2A).

Plotting the data in Figure 2A on a log_10_ scale (Figure 2B) revealed a significant 4.5 log difference in T/V ratios between the drug combination group (ETV + GLP-26) and the control group (no-drug) mice on day 45 (*p* < 0.01). By comparison, the single-therapy groups demonstrated 2.5 log_10_ (GLP-26 only; *p* < 0.005) and 3 log_10_ differences (ETV only; *p* < 0.02) versus non-drug treated, at day 45. These results demonstrated that GLP-26 is active against virus replication in this mouse model. When the tumors were removed at the end of the experiment, no visible necrosis was observed, even when the tumors were sectioned for gross and microscopic examination.

Determining the mechanism of action of GLP-26 on viral replication versus integrated HBV DNA in the AD38 nude mouse xenograft model:

In the nude/AD38 mouse model, virus replication could be initiated from two independent templates. Pre-genomic HBV RNA can be made from integrated HBV DNA under control of the TET repressor, so that once TET was removed from the drinking water, suppression is relieved and pre-genomic RNA synthesis could proceed [14]. However, pre-genomic HBV RNA can also be made from covalently closed circular (ccc) HBV DNA [23]. In either case, pre-genomic HBV RNA would migrate into the cytoplasm, and be packaged into immature core particles along with the virus polymerase. Pre-genomic RNA is then reverse transcribed into virus DNA. In the normal life cycle of the virus, the envelope is acquired by budding, and the resulting mature virus secreted. Among immature cytoplasmic core particles, replication could be blocked by ETV (so that virus DNA synthesis is inhibited) and GLP-26 (which destroys the integrity of immature core particles). When these drugs block the maturation of immature core particles into enveloped virus particles, the partially replicated virus DNA in the cytoplasm re-enters the nucleus and is converted into ccc HBV DNA. Sustained use of these drugs should attenuate this recycling by attacking two independent stages of virus replication. Given that TET inhibits the generation of pre-genomic RNA from integrated HBV DNA, the addition of combination therapy should provide a readout that reflects their effectiveness against virus replication in immature core particles by attenuating virus maturation and by reducing the pool of ccc HBV DNA, both of which should diminish the titer of virus detectable in the serum. Accordingly, mice carrying AD38 tumors were treated with TET from day −10 thru day 0 to suppress virus from integrated HBV DNA (Figure 3). When some groups of mice were taken off TET, and treatment with combination therapy started 10 days later (on day 10; Figure 3A,B), the virus titers in their blood would reflect the ability of combination therapy to prevent the re-emergence of virus from ccc HBV DNA which was made in the absence of TET between days 0 and 10.

When combination therapy (ETV + GLP-26) was administered between days 0 and 35, virus suppression was observed until the combination therapy was withdrawn, and only then some virus rebound was seen, but this did not reach statistical significance (*p* > 0.05) (Figure 3A). This suggests persistence of residual cccDNA which supported virus replication once combination therapy was ended. When mice were treated with the combination therapy between days 0 and 35, without TET reinstatement, T/V increased in six of seven mice, but again this was not statistically significant (*p* > 0.1) (Figure 3B). Therefore, combination therapy partially blocked virus replication regardless of whether pre-genomic RNA originated from integrated or HBV cccDNA, but rebound was observed after removal of ETV/GLP-26.

In contrast, when another group of mice were administered TET between days −10 and 0 and again from day 10 to 45, there was a rapid and significant rebound of T/V in six of seven mice when they were taken off TET during days 0–10 (*p* < 0.02) followed by suppression again when TET was reinstated (Figure 3C). These observations suggest that the lack of rebound in Figure 3A,B resulted from combination therapy, which was absent in Figure 3C during days 0–10. Continuous administration of TET for 55 days consistently produced low T/V ratios (Figure 3D), indicating that virus replication could be suppressed long term from integrated HBV DNA. However, when TET was administered only for days −10 to day 0, T/V significantly increased to day 45 (*p* < 0.001; Figure 3E).

When the T/V ratios from the treated groups were compared to the control group without drug, virus replication was very strong over time in the control group compared to treated mice (Figure 4A).

When the average virus titers for each group were plotted on a log_10_ scale (Figure 4B), and the results compared at day 45, rebound was seen after mice were taken off of combination therapy (curves 3A and 3B) and after TET was removed (curve 3C). In contrast, differences in T/V ratios comparing curves designated 3D and 3E showed a 5.46 log_10_ difference by day 45, demonstrating a combination treatment effect that was robust over more than 5 log. No differences in the body weight or other overt evidence of toxicity were evident in mice administered drug treatments versus those administered placebo regimens as no mice died during the study. In addition, there was no evidence of drug toxicity on the growth of tumor cells among Figure 1, Figure 2, Figure 3 and Figure 4 through 45 days of observation (data now shown), suggesting that the differences in virus titer and tumor volumes were due to antiviral activities of the treatments.

### 3.2. Pharmacokinetic Characterization of GLP-26 in Male Cynomolgus Monkeys

All monkeys were administered GLP-26 without incident and appeared healthy prior to dosing and throughout the duration of the study. The plasma concentration versus time profiles of GLP-26 following IV (1 mg/kg) and PO (5 mg/kg) administration and the in vitro concentrations producing 50% and 90% inhibition of HBV (EC_50_ and EC_90_) are shown in Figure 5A,B, respectively.

The single-dose non-compartment pharmacokinetic parameters of GLP-26 in cynomolgus monkeys are summarized in Table 1.

GLP-26 demonstrated a 33.6% oral bioavailability, with AUC_total_ = 986.6 ± 316.5 and 1660 ± 530 h·ng/mL (mean ± SD) following IV (1 mg/kg) and PO (5 mg/kg) administration, respectively. The terminal t_1/2_ were 0.77 ± 0.23 and 2.44 ± 0.57 for IV and PO doses, respectively.

Plasma binding studies were performed using a 10 µM concentration of GLP-26 (similar to C_max_ following IV administration of 1 mg/kg (Figure 5). Results revealed 86.7% binding in monkey and 89.5% in human plasma, respectively. For comparison, 0.8% of emtricitabine (negative control) and 97.9% of lopinavir (positive control) were bound in human plasma, and 17.9% was bound in 2% FBS (used to measure potency in vitro) [9]., These binding data were used to estimate the in vivo potency from the in vitro EC_50_ (3 nM~1.1 ng/mL) and EC_90_ (30 nM~11 ng/mL) using the equation: in vivo potency = in vitro potency × % bound in plasma/% bound in 2% FBS. The normalized EC_50_ and EC_90_ were superimposed on Figure 5A,B.

### 3.3. Effect of GLP-26 on Human Cardiomyocyte Function

A cardiomyocyte cell culture assay was used to assess the potential for GLP-26-induced cardiotoxicity at 2 and 10 µM (equivalent to the total (bound + free) plasma concentration at C_max_). BMS-986094 was used as a positive control, as this investigational NS5B nucleotide polymerase inhibitor was discontinued for the treatment of hepatitis C, after being linked with cardiac failure in infected persons [24,25]. BMS-986094 was used at 50 µM, as a reference for severe cardiac toxicity for consistency with historical controls by our group. DMSO was used as a negative control. GLP-26 up to 10 µM had no apparent impact on cardiomyocyte function over 48 h compared to negative control as judged by beat rate (Figure 6A), force of contraction and relaxation, (Figure 6B), velocity and contraction to relaxation intervals (Figure 6C). In contrast, with 50 µM of the positive control BMS-986094, cardiomyocyte beating stopped within 2 h and resumed after 8 h with an unstable rhythm.

## 4. Discussion

### 4.1. Effect of Antiviral Monotherapy or Combination Therapy on HBV DNA Copies/mL Serum in Nude Mice Bearing HBV-Infected AD38 Xenografts

The antiviral efficacy of orally administered GLP-26 (60 mg/kg) and/or ETV (10 mg/kg) administered three times per week for up to 45 days was studied in nude mice bearing HBV-infected AD38 tumor xenografts. Although efforts were made to ensure reproducibility, tumor sizes and viral titers varied within each group, and larger tumors correlated with higher viral titers. These variations may be due to differences in tumor seeding which could result from slight differences in the depth of injection, and/or variations in each tumor’s blood supply [26,27]. Therefore, viral titers were normalized by tumor volume (T/V) before analysis. GLP-26 had considerable antiviral activity when combined with ETV (Figure 1C). Although the physiology of AD38 tumor xenografts differ from that of infected human liver, the effect of GLP-26 and ETV on serum HBV titers observed with the tumor xenograft model resembled those observed in our previous study in a humanized mouse model of HBV infection [9].

### 4.2. Determining the Mechanism of Action of GLP-26 on Viral Replication Versus Integrated HBV DNA in the AD38 Nude Mouse Xenograft Model

Pre-genomic HBV RNA could be transcribed from both integrated and cccDNA in the AD38 model, although pre-genomic RNA is transcribed only from cccDNA among infected patients [23]. Continuous exposure to TET strongly inhibited virus replication (Figure 3D), but not to undetectable levels, suggesting that the shut-down of pre-genomic HBV RNA production is incomplete under these experimental conditions. In contrast, the slow decay in T/V observed upon exposure to TET from day 10 to 45 (Figure 3C) may reflect the stability and/or recycling of HBV DNA from immature core particles back into the nuclei of cells, resulting in replenishment of the template for pre-genomic RNA synthesis.

Experiments using combination therapy with or without TET also provided information regarding the contribution of integrated and HBV cccDNA templates to virus replication (Figure 3). Virus rebound after the termination of combination therapy may reflect the continued production of low levels of pre-genomic RNA from integrated HBV DNA only (curve 3A in Figure 4B) or from both templates (curve 3B in Figure 4B). Comparison of curves 3A and 3B (combination therapy) to curves 3D and 3E (TET only) in Figure 4B, however, shows that GLP-26 is an effective antiviral agent, especially when combined with ETV. However, long-term administration of this drug combination may be required to maintain persistent suppression of HBV replication.

### 4.3. PK Studies in Cynomolgus Monkeys

No signs of toxicity were observed in cynomolgus monkeys administered single IV or PO doses of GLP-26. However, the experiments were not designed to detect signs of metabolic or histological toxicity. Similarly, GLP-26 was safely administered to humanized mice at 60 mg/kg per day for 10 weeks of treatment (PO) [9]. Approximately 34% of the 5 mg/kg oral dose of GLP-26 was absorbed in cynomolgus monkeys, producing a C_max_ equal to 380.7 ng/mL (which is~30 times the in vitro EC_50_ versus HBV) and the terminal t_1/2_ was 2.44 h. The good oral bioavailability of GLP-26 and prolonged half-life after oral administration observed in cynomolgus monkeys were in agreement with our earlier study in CD-1 mice [9]. In that study, mice were administered GLP-26 via PO (30 mg/kg) or IV (15 mg/kg) routes and demonstrated 61% PO bioavailability, and an increase in terminal t_1/2_ from 1.6 h to > 6 h following IV and PO administration, respectively. The 3.17 h mean input time (MIT = MRT_PO_ − MRT_IV_) computed for cynomolgus monkeys suggests that the prolonged t_1/2_ following PO administration may result from a slow absorption into the plasma compartment. A single 5 mg/kg PO dose of GLP-26 produced a plasma concentration which remained above the in vitro EC_90_ (corrected for plasma binding) for between 5 and 12 h, suggesting that GLP-26 may need to be administered more than once a day. However, cynomolgus monkeys (species bodyweight~3 kg) are considerably smaller than humans (~70 kg) and larger species generally metabolize drugs slower. Therefore, PK predictions based on preclinical data need to be confirmed in early phase 1 trials in humans [28]. The AUC_total_ observed in the cynomolgus monkeys administered 5 mg/kg GLP-26 orally was similar to AUC_0–7 h_ observed in HBV-infected humanized CD-1 mice successfully treated with GLP-26 and ETV [9].

### 4.4. Effect of GLP-26 on Human Cardiomyocyte Function

Cardiotoxicity is a critical limiting side effect for drug development, including for antiviral drugs [12,13,23,24]. Here, we confirmed that GLP-26 does not demonstrate any effect on cardiomyocyte function at in vitro concentrations up to 10 µM (considerably higher in plasma considering >86% binding [16]).

### 4.5. Overall Conclusions

The lack of toxicity and potent anti-HBV potency in vitro, efficacy in murine models of HBV infection, and favorable PO pharmacokinetics in rodents and cynomolgus monkeys suggest potential for further preclinical development of GLP-26.

## Figures and Tables

**Figure 1 viruses-13-00114-f001:**
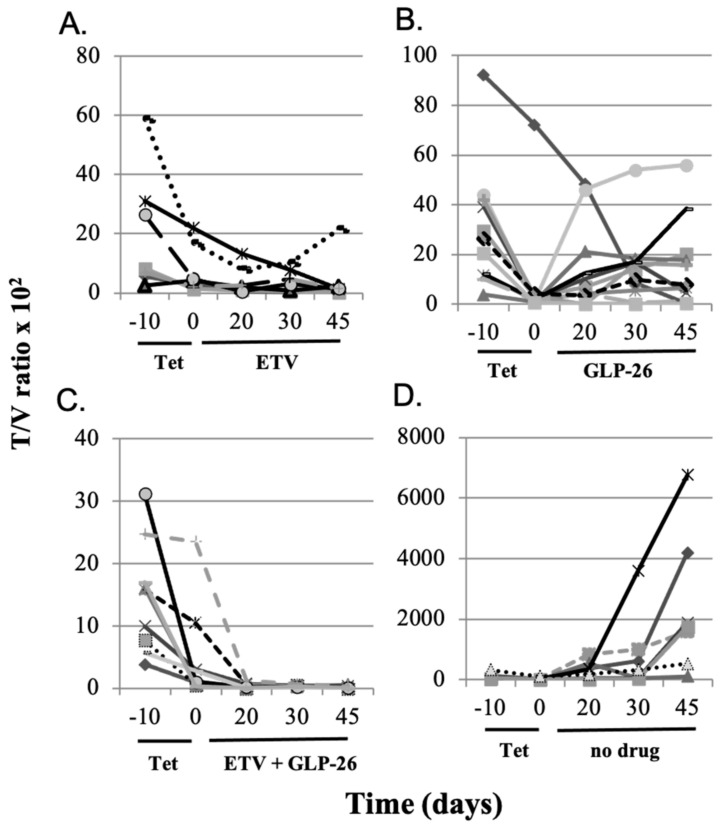
HBV titers (DNA copies/mL serum per mm^3^ tumor (T/V ratios) × 10^2^) of individual nude mice bearing HBV-infected AD38 xenografts. Mice in all groups were given TET in their drinking water from day minus 10 to day 0. On day 0, TET was replaced with the following treatments: (**A**) continuous ETV from day 0 to day 45, (**B**) continuous GLP-26 from day 0 to day 45, (**C**) continuous ETV + GLP-26 from day 0 to day 45, and (**D**) no drug was provided from day 0 to day 45.

**Figure 2 viruses-13-00114-f002:**
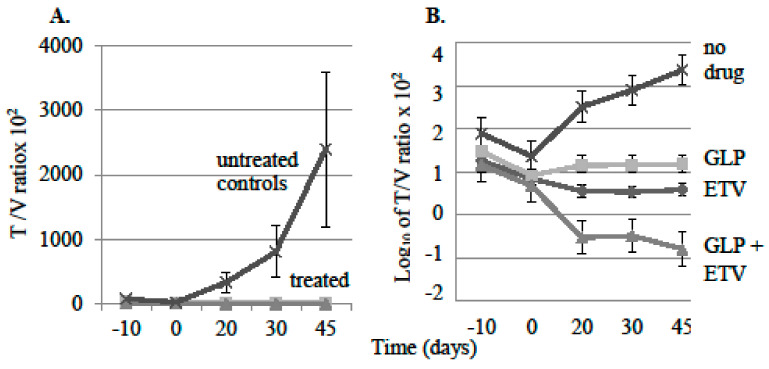
(**A**) Average HBV titer (DNA copies/mL serum per mm^3^ tumor (T/V ratios) × 10^2^, of treated and untreated groups of mice. Error bars were calculated as standard error of the means (SE). (**B**) Data from panel A plotted logarithmically for each group of mice.

**Figure 3 viruses-13-00114-f003:**
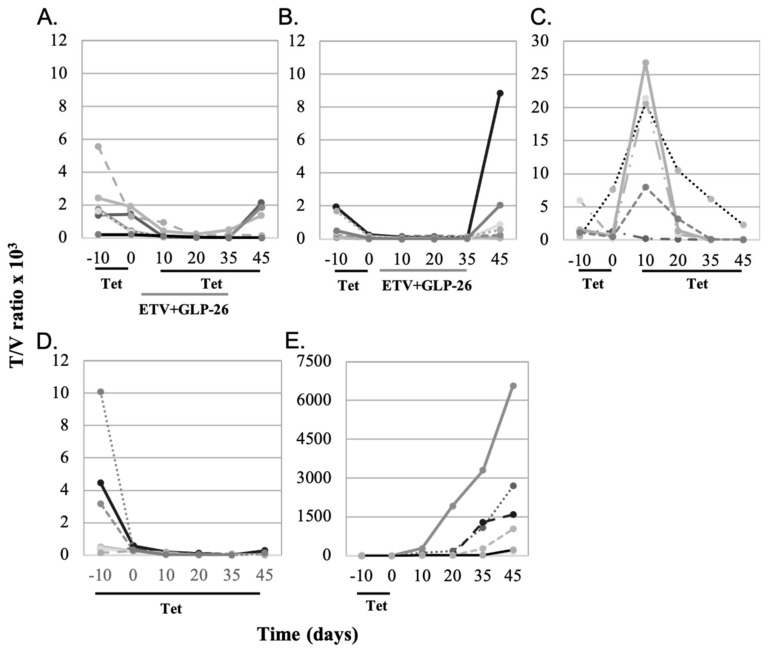
(**A**) Mice were treated with TET from day −10 to 0, taken off TET for days 0–10 and then put back on TET for days 10–45. They were also treated with combination therapy (ETV + GLP-26) from day 0 to 35. (**B**) Mice were treated with TET from day −10 to 0, and then put on combination therapy from day 0 to 35. Mice were not treated from day 35 to 45. (**C**) Mice were treated with TET from day −10 to 0, taken off TET for days 0–10 and then put back on TET for days 10–45. (**D**) Mice were treated with TET from day −10 to 45. (**E**) Mice were put on TET from day −10 to 0, and not treated further.

**Figure 4 viruses-13-00114-f004:**
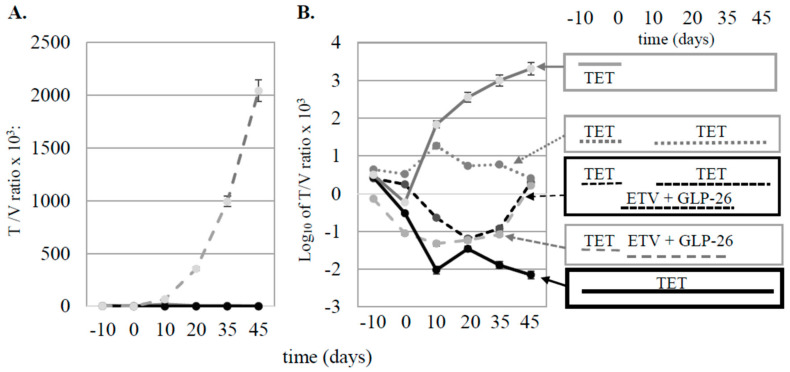
(**A**) Average ratios of T/V for treated (solid black line) and untreated (dashed gray line) groups of mice. (**B**) Data from panel A plotted logarithmically for each group of mice. The treatment protocols for each group of mice are indicated in the rectangles to the right of panel B, and correlate with the groups in Figure 3 as follows: 3A (dashed black line), 3B (dashed gray line), 3C (dotted grey line), 3D (solid black line), and 3E (solid gray line).

**Figure 5 viruses-13-00114-f005:**
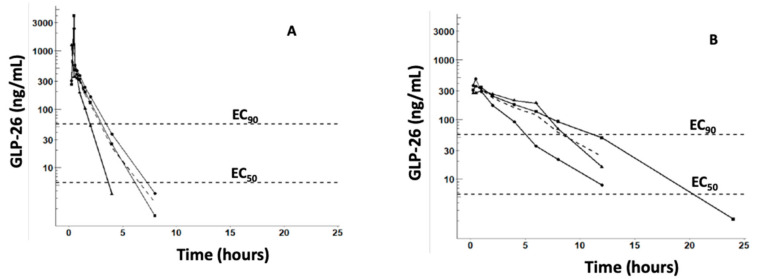
Plasma concentration versus time profiles of GLP-26 in cynomolgus monkeys administered 1 mg/kg IV (**A**) or 5 mg/kg PO (**B**). Solid curves and markers represent concentrations of GLP-26 in individual monkeys, while dashed curves represent means. Only concentrations >limit of quantification (1 ng/mL), are plotted. Also plotted are the in vitro EC_50_ (3 nM~1.1 ng/mL) and EC_90_ (30 nM~11 ng/mL) versus HBV, corrected 86.7% binding in monkey plasma and 17.9% binding in vitro (2% FBS), respectively.

**Figure 6 viruses-13-00114-f006:**
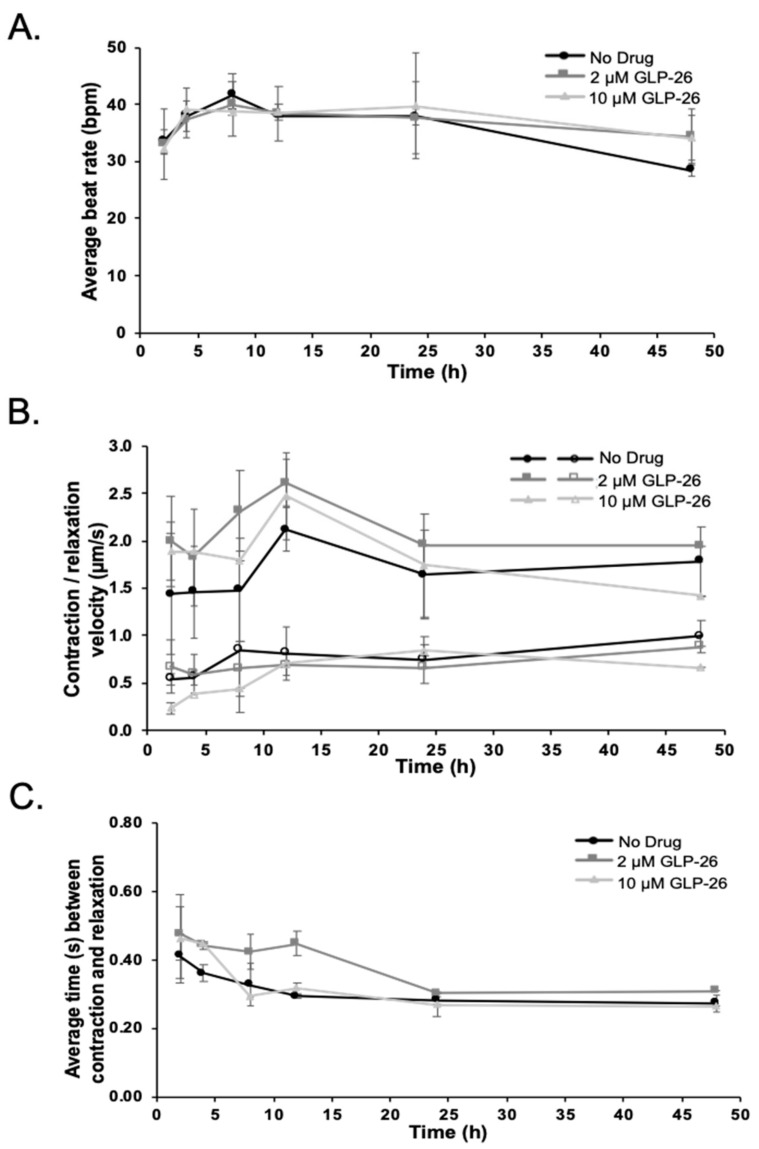
Cardiomyocytes treated with 2 or 10 µM of GLP-26. (**A**) Beat rate. (**B**) Force of cardiomyocytes contraction and relaxation (solid markers for contraction, open markers for relaxation). (**C**) Contraction to relaxation time intervals.

**Table 1 viruses-13-00114-t001:** PK of GLP-26 administered by IV infusion (1 mg/kg over 0.5 h) or PO (5 mg/kg) to male cynomolgus monkeys. N = number of monkeys; C_max_ and T_max_ are maximal observed plasma concentration and the time it was observed, respectively; AUC_total_ = area under the plasma concentration versus time curve (extrapolated); CL/F = dose/AUC_total_, where F is the fraction of the dose systemically absorbed (=1 for IV administration); t1/2 = terminal first-order half-life; AUMC = area under the first moment curve; MRT = mean residence time; Vss = steady-state volume of distribution; SD = standard deviation; %CV = percent coefficient of variations. NA = not applicable. The percent of GLP-26 bound in the plasma of cynomolgus monkeys was 86.7%.

	1 mg/kg IV 1 h Infusion			5 mg/kg, PO		
	Mean (*n* = 3)	SD	% CV	Mean (*n* = 3)	SD	% CV
C_max_, ng·mL^−1^	2513	1339	53.3	380.7	88.21	23.2
T_max_, h	0.40	0.13	32.9	0.67	0.29	43.30
AUC_total_, ng·h·mL^−1^	986.6	316.42	32.1	1660	530.05	31.9
AUMC, ng·mL^−1^·h^2^	1058	525.5	49.7	7435	4123	56.5
t_1/2_, h	0.77	0.22	29.1	2.44	0.57	23.2
MRT, h	1.2	0.27	26.6	4.19	1.38	33.0
CL, L·h·kg^−1^	1.106	0.43	39.1	3,285	1.28	39.1
V_ss_, Lkg^−1^	1.065	0.18	16.6	NA	NA	NA

## Data Availability

The data presented in this study are available on request from the corresponding author.

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
