# Peer review of "Studies on the Efficacy, Potential Cardiotoxicity and Monkey Pharmacokinetics of GLP-26 as a Potent Hepatitis B Virus Capsid Assembly Modulator"

_viruses, 2021, doi:10.3390/v13010114_

Round 1

Reviewer 1 Report

This manuscript presents overall excellent work on GLP-26, a very important HBV CpAM. The work is centered around in vivo efficacy in an HBV nude mouse model with

 HBV infected AD38 xenografts and in vivo PK with male cynomolgus monkeys. In addition, cardiotoxicity was assessed in primary human cardiomyocytes. Overall, the efficacy data strongly indicate that GLP-26 can effectively reduce viral load, especially when used in combination with ETV. The PK profile is largely favorable with an oral bioavailability of 34%. Importantly, at 10 uM GLP-26 had no apparent impact on cardiomyocytes functions. The paper is very well written and all data are clearly presented. Only a few very minor issues are identified for revision consideration.

  • Line 185: experimental description and data are missing for the combination therapy stopped on day 35.
  • For section 3.3 (the effect on cardiomyocytes): a) what about the effect of ETV / GLP-26 combination, which seemed much more effective in the efficacy assay? Combination effects like this can be considered for future work. b) line 301: data for BMS-986094 are missing.
  • Typos: a) Figure 1 &2 legend, “per mm3 tumor (T/V ratios) x 102”, should 3 and 2 be superscripted? b) Figure 6 legend, concentration unit is unrecognizable symbols; c) same for line 354.

Author Response

The paper is very well-written and all data are clearly presented. Only a few very minor issues are identified for revision consideration.

Line 185: experimental description and data are missing for the combination therapy stopped on day 35.

  • To clarify, you must mean day 45.  The data for Fig. 1C is already provided in line 179 and for Fig. 1D in line 187. The experimental description of when mice are on and off various treatments is indicated under each of the respective graphs in the figure.

For section 3.3 (the effect on cardiomyocytes): a) what about the effect of ETV / GLP-26 combination, which seemed much more effective in the efficacy assay? Combination effects like this can be considered for future work. b) line 301: data for BMS-986094 are missing. 

  • Unfortunately, we did not test the effect of ETV/GLP-26 combination on cardiomyocytes. But as the reviewer suggested, this will definitely be considered for future work. We moved the statement “with 50 µM BMS-986094 the positive control, cardiomyocyte beating stopped within 2 h and resumed after 8 hr with an unstable rhythm” up just after the effect of GLP-26. Since the beating stopped at the very first time point for BMS-986094, this is a sign that the positive control worked as expected.

Typos: a) Figure 1 &2 legend, “per mm3 tumor (T/V ratios) x 102”, should 3 and 2 be superscripted? b) Figure 6 legend, concentration unit is unrecognizable symbols; c) same for line 354. 

  • As suggested, we have made the changes, and now updated the revised manuscript.

Reviewer 2 Report

Hurwitz and colleagues present their studies to detail the possible application of GLP-26 to HBV therapy in a combination regiment with entecavir (ETV) for HBV infections.

Here, they present HBV transfected AD38 xenographs in a nude mouse model to study the effects on viral replication with the combination (or simple usage) of anti-HBV chemical compounds.

This system and study is relevant, as it might have a positive impact on the lives of people infected with HBV. It is my opinion, that the study is interesting, but could have some improvements in certain technical aspects of the experimental design and text.

Major points regarding experimental design, results, description of the data and discussion:

- Authors do not report the known toxicity of NUCs and interferon therapies, or relate it in any way to tumour growth.

- Authors present all the data as T/V ratio, and correctly report they observeddifferent tumour sizes in the discussion.
It is my opinion that it is also crucial to add the data on tumour sizes for the different animals on all reported data/time points, so that we can ascertain a possible effect from time, Tet, ETC, GLP-26, and their combinations on the xenographs. Authors report in line 158 that larger tumours have higher HBV titers, it would be of importance to know if the T/V ratio is independent or dependent of such size.

- In the whole study, the HBV levels were measured by RT qPCR, which the authors report that it can be counfounded between cccDNA and viral pre-genomic RNA. Moreoever, larger xenographs should produce higher amounts of viruses (which is correctly dealed with, with the T/V ratio), but could still affect the sensitivity and specificity of the system.

It would be interesting to show viral proteins levels (or antibody production in the mice against the viruses as an alternative), at least, in some of the time points or initial data-sets.

If the xenograph cells would be dying over time (or affected by the chemical compounds) this could affect the conclusions of the study significantly.

-In the results presentation of figure 3A and B the manuscript refers to the rebound happening after ETV+GLP-26 +- Tetracycline is stopped at day 35, as differences between pre-genomic are explained between lines 202 and 212.

  • The authors do not explain or mention to why such rebound is not present in the conditions of Fig3C (only Tet) that is comparable to Fig3A. Such rebound is not present and should be more prominent in Fig3C while comparing to Fig3A. One would expect that the ETV+GLP-26 inhibition of DNA proliferation would limit such rebound compared to Fig 3C where tet was removed between days 0 and 10, as for the Fig 3A conditions, where overall viral RNA production should be higher.
  • The reasoning between lines 202 and 212 should be more complete and more accessible to a non specialist reader.

- The GLP-26 pharmokinetics are shown in Fig5 but authors do not discuss what would be the impact of such discoveries into a human patients’ application, as it is shown, GLP-26 should be administered several times a day for a correct action. If so, it would also be important to discuss that further modifications into GLP-26 could be benefitial for a translational application to HBV patients.

- Fig 6 B has open and closed boxes, it is not stated what the data sets are.

- Authors refer using BMS-986094 as a positive control for toxicity but no data was presented for the reader to compare to GLP-26. Also, no statistical analysis was reported to such control.

- Authors should show both the time-lapse imaging data and the tracking analysis on section 3.3.

- Authors should include in the discussion certain caveats of the system, such as what are the downsides of only measuring DNA by reverse transcriptase qPCR, the time relation to pharmokinetics of GLP-26 to a possible therapy.

- The y axis on figures 1 and 3 have differences of a lot of orders of magnitude, I understand that the T/V ratios vary considerably, but it becomes very hard to compare the data (for example Fig 3 A and Fig 3E, and Fig1A and D; especially in the T/V x 103 ratios between 0 and 1, and to compare the initial level of replication between days -10 and 0 between the conditions.

- Authors do not report what the error bars represent (SEM, STDEV?). In Fig 2A, error bars only go to the positive side while others fluctuate to lower and higher errors.

- Does the data from fig 1 correspond to the data plotted in Fig2? I would expect that the error bars would go higher in the Fig2.B GLP+ as in Fig1 B there is a dataset (dark gray) that shows a outlier behavior with higher T/V ratio if STDEVs are shown as error bars.

- t-test should be used only to compare two discrete conditions, authors used t-tests to compare different time points (it should be stated more clearly if it’s always the same time points comparison). If the comparisons are between different time-points pairs, other statistical tests should be used.

t-tests should be used only after showing gaussian distribution of data, it was not reported or shown that such tests were performed, but the low number of data points and variability, sugest that it will not be a normal distribution, therefore t-tests should not be used.

Major points about the manuscript writing, clarity, and presentation of data

- The abstract contains a high number of acronyms, it would be to the interest of a larger audience to fully describe the system using less acronymes, especially for the specific terms relatable to this article and mouse model (Tmax, Cmax, AUCtotal, po, etc).  

- The abstract reports a HBV model of HBV infected AD38 xenografts while technically they are stably transfected cells, as later correctly described in the material and methods and introduction.

- In the introduction, the authors present the AD38 xenograft system where the expression of pre-genomic RNA is repressed in the presence of Tetracycline. For the clarity of the reader, authors should detail the system better in the introduction and results, even though it has been used in previously well cited articles. It is a Tet-off tTA system as shown in Gossen et al PNAS 1992). An inclusion of a graphic detailing of the system would be good to have.

 As examples of the lacking clarification of the Tet-off:

  "[...]with HBV  and  under  the  control  of  a  tetracycline  (TET)  operon  were  used  [10,  11]";

"[...]pre-genomic  HBV  RNA  can  be  made  from  integrated  HBV  DNA  under control  of  the  TET  repressor  once  TET  was  removed  from  the  drinking  water [...]"

  only in line 321 the authors then describe that Tetracycline presence inhibits virus replication :

"Continuous exposure to TET strongly inhibited virus replication (Figure 3D)[...]"

- Authors refer to Feitelson, et al. and Schinazi, et al. when presenting the AD38 TET system, but do not reference Ladner  et  al.,1997, where the system was initially described. Such publication was then mentioned in the material and methods section (citation #14), it would be important to reference it appropriately in the introduction.

- Line 220 to 220. Authors show a comparison of statistical significance between ETV+ GLP-26, it is unclear what the comparison was, if another data set, or different time points in the ETV+ GLP-26 dataset.

Material and methods

- The carriers in the chemicals section were not detailed for all used components (water, DMSO, etc).

- The primers for RT-qPCR or the system were not detailed.

- Line 150: Authors used computation motion tracking software, but there's no detail on which one, or any data on the imaging or the data analysis that is pertinent to it.

 Typos and other minor details:

 Line 28: "add-on to combination treatment" is a redundant term.

 Figure 1 legend. Authors report "continuous treatments", please consider using another term, as it was an oral gavage at Mondays, Wednesdays, and Fridays and not a continuous system (permanent IV for example).

Line 202: "Theoretically"

Fig 4B: it would be easier for a reader to have the 5 different data sets labelled as in Fig 3 and not 3A to E.

Author Response

Responses to Reviewers: Manuscript ID: viruses-1038675

Studies on the efficacy, potential cardiotoxicity and monkey pharmacokinetics of GLP-26 as a potent hepatitis B virus Capsid assembly modulator

Selwyn J. Hurwitz, Noreen McBrearty, Alla Arzumanyan, Eugene Bichenkov, Sijia Tao, Leda Bassit, Zhe Chen, James Kohler, Franck Amblard, Mark Feitelson, Raymond F. Schinazi *

We thank the Reviewers for overall positive review and helpful suggestions and for the opportunity to improve the manuscript.

Reviewer #2: This system and study is relevant, as it might have a positive impact on the lives of people infected with HBV. It is my opinion, that the study is interesting, but could have some improvements in certain technical aspects of the experimental design and text.

Major points regarding experimental design, results, description of the data and discussion:

- Authors do not report the known toxicity of NUCs and interferon therapies, or relate it in any way to tumour growth.

Unlike nucleosides developed for cancer chemotherapy (e.g., Gemcitabine), Entecavir and most selective nucleosides developed as antiviral agents have minimal cytotoxicity, at clinically relevant concentrations or doses. Therefore, it is unlikely that these FDA approved antiviral nucleoside analogs would inhibit tumor growth, at the in vivo concentrations used in the study. This was in fact demonstrated in the study, as outlined in more detail in our responses below. 

(Hurwitz, S.J. & Schinazi, R.F.: Practical considerations for developing nucleoside reverse transcriptase inhibitors. Drug Discovery Today: Technologies. Fall:e183-e193, 2012. Hurwitz, S.J., Schinazi, R.F.: Prodrug strategies for improved efficacy of nucleoside antiviral inhibitors. Current Opinion in HIV and AIDS ;8(6):556-64. 2013). Likewise, GLP-26 has negligible cytotoxicity (Amblard F, Boucle S, Bassit L, Cox B, Sari O, Tao S, Chen Z, Ozturk T, Verma K, Russell O, Rat V, de Rocquigny H, Fiquet O, Boussand M, Di Santo J, Strick-Marchand H, Schinazi RF. 2020. Novel hepatitis B virus capsid assembly modulator induces potent antiviral responses in vitro and in humanized mice. Antimicrob Agents Chemother 64(9): e01351-20. PMCID: PMC698570).

- Authors present all the data as T/V ratio, and correctly report they observed different tumour sizes in the discussion. It is my opinion that it is also crucial to add the data on tumour sizes for the different animals on all reported data/time points, so that we can ascertain a possible effect from time, Tet, ETC, GLP-26, and their combinations on the xenographs. Authors report in line 158 that larger tumours have higher HBV titers, it would be of importance to know if the T/V ratio is independent or dependent of such size. 

Tumor sizes depend upon how many cells “take” after sub-cutanious injection, even if no drugs are used in the system.  This is normal biological variability, even when the same number of cells are injected into mice of the same age and gender.  This why at day minus 10, each mouse starts with a different baseline of virus, which serves as a control for subsequent treatment of each mouse. As already stated in the manuscript, the T/V ratios increase with tumor size.  There is no impact of treatments upon tumor growth over time.  This is indicated by an additional sentence at line 252 in the manuscript.  

 - In the whole study, the HBV levels were measured by RT qPCR, which the authors report that it can be counfounded between cccDNA and viral pre-genomic RNA. Moreoever, larger xenographs should produce higher amounts of viruses (which is correctly dealed with, with the T/V ratio), but could still affect the sensitivity and specificity of the system. 

The sensitivity and specificity of the measurements is determined by the precision of PCR, which demonstrate both characteristics.  This reproducible precision and quantification is central to discerning differences in virus levels under different conditions, which in turn is important for suggesting the origin of virus replication in each of the experimental conditions.

It would be interesting to show viral proteins levels (or antibody production in the mice against the viruses as an alternative), at least, in some of the time points or initial data-sets.

The experiments were done in nude mice, which do not produce antibodies, since there is no T helper function for B cells.  As to measuring virus protein levels, this is much harder to quantify compared to the real time PCR measurements of virus genome equivalents in serum, which is why the latter is reported instead. For western blots, the best we can do is report fold change in treated compared to untreated, but for PCR we report the actual number of virus genome equivalents.  Protein analysis would not augment or change the results, since in this system, all the virus proteins made follow virus replication.

If the xenograph cells would be dying over time (or affected by the chemical compounds) this could affect the conclusions of the study significantly. 

When the tumors were removed at the end of the experiment, no visible necrosis was observed, even when the tumors were sectioned for gross and microscopic examination.  We terminated the experiments at 45 days, because if the experiment was extended, necrosis and tumor ulceration would have occurred.  So, this was taken into consideration in the original experimental design. This is now included in line 94, 205 and 275 of the revised manuscript.

-In the results presentation of figure 3A and B the manuscript refers to the rebound happening after ETV+GLP-26 +- Tetracycline is stopped at day 35, as differences between pre-genomic are explained between lines 202 and 212.

The authors do not explain or mention to why such rebound is not present in the conditions of Fig3C (only Tet) that is comparable to Fig3A. Such rebound is not present and should be more prominent in Fig3C while comparing to Fig3A. One would expect that the ETV+GLP-26 inhibition of DNA proliferation would limit such rebound compared to Fig 3C where tet was removed between days 0 and 10, as for the Fig 3A conditions, where overall viral RNA production should be higher. 

In Fig. 3A, there was no rebound during days 0-10 because the mice were treated with combination therapy.  The latter was absent in Figure 3C.  A comment is added for clarification on line 244 of the manuscript.

The reasoning between lines 202 and 212 should be more complete and more accessible to a non-specialist reader. 

This paragraph has been mostly re-written to provide a better understanding for the non-specialist.

  • The GLP-26 pharmacokinetics are shown in Fig 5 but authors do not discuss what would be the impact of such discoveries into a human patients’ application, as it is shown, GLP-26 should be administered several times a day for a correct action. If so, it would also be important to discuss that further modifications into GLP-26 could be benefitial for a translational application to HBV patients.

The pharmacokinetics in monkeys suggest that more than one dose per day may be needed. However, cynomolgus monkeys are considerably smaller (~ 3 kg) than humans (~ 70 kg) and larger species generally metabolize drugs slower than smaller ones. Although preclinical studies may give some indication of pharmacokinetics and humans, extrapolating pharmacokinetics between species needs to be confirmed in phase 1 studies. This has been included in line 375 of the revised manuscript, and references 28 concerning extrapolation of PK between species were included.

- Fig 6 B has open and closed boxes, it is not stated what the data sets are.

- Authors refer using BMS-986094 as a positive control for toxicity but no data was presented for the reader to compare to GLP-26. Also, no statistical analysis was reported to such control.

- Authors should show both the time-lapse imaging data and the tracking analysis on section

We have added in in the figure legend that solid markers are for contractions and open markers for relaxation. We stated in text that with 50 µM BMS-986094 the positive control, cardiomyocyte beating stopped within 2 h and resumed after 8 hr with an unstable rhythm. Beating stopped at the very first time point for BMS-986094 is a clear sign that the positive control worked well.

To clarify, data are shown in figures. There’s no specific imaging, but videos outputted by the software were saved on computer (which are not included in the paper).

- Authors should include in the discussion certain caveats of the system, such as what are the downsides of only measuring DNA by reverse transcriptase qPCR, the time relation to pharmacokinetics of GLP-26 to a possible therapy. 

qPCR performed on samples from each mouse at the various time points, were as performed in triplicate, and a standard curve was generated for each experimental run. qPCR is precise and reproducible.  Samples from test and negative control were evaluated at the same time to minimize experimental variability.  Virus protein expression is much less precise, and as in human infections, does not always closely correlate with virus replication. Virus replication was the preferred endpoint in this study.

Figure 5 superimposes the single-dose pharmacokinetics of GLP-26 in cynomolgus monkeys over the in vitro anti-HBV potencies (EC50 and EC90). As discussed in the text, this indicates the duration of expected antiviral activity for a single oral or IV dose of compound. Since the pharmacokinetics were measured in cynomolgus monkeys and the antiviral efficacy studies were conducted in nude mice it was not feasible to directly link the pharmacokinetics and efficacy studies, as proposed by this reviewer.

 - The y axis on figures 1 and 3 have differences of a lot of orders of magnitude, I understand that the T/V ratios vary considerably, but it becomes very hard to compare the data (for example Fig 3 A and Fig 3E, and Fig1A and D; especially in the T/V x 103 ratios between 0 and 1, and to compare the initial level of replication between days -10 and 0 between the conditions.

Scales where different in Fig. 1D and Fig. 3E compared to the other panels in the respective figures, because without consistent treatment with TET, and/or single or combination therapy, the virus titers could get to very high levels in the blood of these mice. Therefore, a more direct comparison of these differences is shown in Fig. 4, to demonstrate how strongly the drugs suppress virus replication and accumulation in the blood plasma.

- Authors do not report what the error bars represent (SEM, STDEV?). In Fig 2A, error bars only go to the positive side while others fluctuate to lower and higher errors.

Figure 2 has now includes error bars in both directions and represents standard errors of the mean.  This has been added in the figure legend (line 195)

- Does the data from Fig 1 correspond to the data plotted in Fig 2? I would expect that the error bars would go higher in the Fig 2.B GLP+ as in Fig1 B there is a dataset (dark gray) that shows a outlier behavior with higher T/V ratio if STDEVs are shown as error bars. 

Yes. Error bars were plotted as calculated.

 - t-test should be used only to compare two discrete conditions, authors used t-tests to compare different time points (it should be stated more clearly if it’s always the same time points comparison). If the comparisons are between different time-points pairs, other statistical tests should be used. 

We compared two discrete conditions, and the two conditions are separate time points.

t-tests should be used only after showing gaussian distribution of data, it was not reported or shown that such tests were performed, but the low number of data points and variability, suggest that it will not be a normal distribution, therefore t-tests should not be used. 

As mentioned by this reviewer, the data set was small. Therefore, it was not feasible to prove that the data set followed a Gaussian distribution. However, comparisons were made using discrete intervals. Given the limitations of the data, a t-test was used. Data were obtained from the same experiments, using the nude mouse-HBV-AD38 tumor model system, so t-tests were performed assuming equal variances.

Major points about the manuscript writing, clarity, and presentation of data

- The abstract contains a high number of acronyms, it would be to the interest of a larger audience to fully describe the system using less acronymes, especially for the specific terms relatable to this article and mouse model (Tmax, Cmax, AUCtotal, po, etc).  

We have replaced the PK acronyms in the abstract, as suggested by the reviewer.

 - The abstract reports a HBV model of HBV infected AD38 xenografts while technically they are stably transfected cells, as later correctly described in the material and methods and introduction. 

Yes, in line 19, “infected” is changed to “transfected” which more accurately describes the system used.

- In the introduction, the authors present the AD38 xenograft system where the expression of pre-genomic RNA is repressed in the presence of Tetracycline. For the clarity of the reader, authors should detail the system better in the introduction and results, even though it has been used in previously well cited articles. It is a Tet-off tTA system as shown in Gossen et al PNAS 1992). An inclusion of a graphic detailing of the system would be good to have. 

Modifications were added to better outline the system used in lines 56-63. This nouse AD38 system was developed by us and we have published extensively on that model. Relevant references on this mouse model are mentioned in the paper.

 - Authors refer to Feitelson, et al. and Schinazi, et al. when presenting the AD38 TET system, but do not reference Ladner et  al.,1997, where the system was initially described. Such publication was then mentioned in the material and methods section (citation #14), it would be important to reference it appropriately in the introduction. 

This citation was added to line 57 and the references renumbered accordingly.

 - Line 220 to 220. Authors show a comparison of statistical significance between ETV+ GLP-26, it is unclear what the comparison was, if another data set, or different time points in the ETV+ GLP-26 dataset. 

This paragraph was rewritten to clarify this and other details, as suggested above by this reviewer.

 Material and methods

- The carriers in the chemicals section were not detailed for all used components (water, DMSO, etc).

- The primers for RT-qPCR or the system were not detailed. 

qPCR was carried out exactly as described earlier [10].  This was added to line 85 in section 2.2.

- Line 150: Authors used computation motion tracking software, but there's no detail on which one, or any data on the imaging or the data analysis that is pertinent to it.

To clarify the computational motion tracking software protocol was referenced (#22) and we made no modifications from the original protocol.

  Typos and other minor details:

 Line 28: "add-on to combination treatment" is a redundant term. 

The word “combination” is deleted.

We propose using GLP-26 as an “add-on” to the existing antiviral regimens and not as a replacement of existing drugs in a regimen.

 Figure 1 legend. Authors report "continuous treatments", please consider using another term, as it was an oral gavage at Mondays, Wednesdays, and Fridays and not a continuous system (permanent IV for example). 

Mice get continuous treatments because drugs are added to their drinking water; the gavage is for the monkey experiments.

Line 202: "Theoretically"

This is part of the paragraph that was rewritten, and for clarification, this word no longer appears in the manuscript.

 Fig 4B: it would be easier for a reader to have the 5 different data sets labelled as in Fig 3 and not 3A to E.

The treatment protocols for each group of mice are added to the righthand side of Fig. 3B for clarity.
